# Model-based assessment of Chikungunya and O'nyong-nyong virus circulation in Mali in a serological cross-reactivity context

Nathanaël Hozé [1,6], Issa Diarra[2,3,6], Abdoul Karim Sangaré[3,4], Boris Pastorino[2], Laura Pezzi [2,5], Bourèma Kouriba[3,4], Issaka Sagara[3], Abdoulaye Dabo[3], Abdoulaye Djimdé [3], Mahamadou Ali Thera [3], Ogobara K. Doumbo·[7], Xavier de Lamballerie [2,8✉] & Simon Cauchemez [1,8]

Serological surveys are essential to quantify immunity in a population but serological cross-reactivity often impairs estimates of the seroprevalence. Here, we show that modeling helps addressing this key challenge by considering the important cross-reactivity between Chikungunya (CHIKV) and O'nyong-nyong virus (ONNV) as a case study. We develop a statistical model to assess the epidemiology of these viruses in Mali. We additionally calibrate the model with paired virus neutralization titers in the French West Indies, a region with known CHIKV circulation but no ONNV. In Mali, the model estimate of ONNV and CHIKV prevalence is 30% and 13%, respectively, versus 27% and 2% in non-adjusted estimates. While a CHIKV infection induces an ONNV response in 80% of cases, an ONNV infection leads to a cross-reactive CHIKV response in only 22% of cases. Our study shows the importance of conducting serological assays on multiple cross-reactive pathogens to estimate levels of virus circulation.

[1] Mathematical Modelling of Infectious Diseases Unit, Institut Pasteur, Université de Paris, UMR2000, CNRS, Paris, France. [2] Unité des Virus Émergents, (UVE: Aix-Marseille Univ-IRD 190-INSERM 1207-IHU Méditerranée Infection), Marseille, France. [3] Malaria Research and Training Center, USTT, Bamako, Mali. [4] Centre d'Infectiologie Charles Mérieux, Bamako, Mali. [5] EA7310, Laboratoire de Virologie, Université de Corse-Inserm, Corte, France. [6] These authors contributed equally: Nathanaël Hozé and Issa Diarra. [7] Deceased: Ogobara K. Doumbo. [8] These authors jointly supervised this work: Xavier de Lamballerie and Simon Cauchemez. ✉email: xavier.de-lamballerie@univ-amu.fr

Mosquito-borne arboviruses cause millions of infections worldwide. The recent emergence of Zika and Chikungunya virus (CHIKV) in the Americas highlights the evolving threat that arboviruses pose to human health[1]. Human activities that disrupt ecosystems increase the risk of adaptation of arboviruses from a sylvatic cycle (where transmission occurs mainly in wild animals, and humans are a dead-end host) to a domestic cycle (where vector-mediated transmission occurs among humans)[2]. Moreover, these viruses have the ability to spread across regions and to cause widespread epidemics[3].

It is therefore important to closely monitor arboviruses with a potential for emergence and to evaluate the associated risk for global emergence. Consider for example O'nyong-nyong virus (ONNV), an alphavirus mainly transmitted by Anopheles funestus and Anopheles gambiae and known only in Africa. Although large outbreaks have occurred in the past (2 million cases in East and West Africa in 1959–1962[4,5]), we only have limited knowledge of its current level of circulation in Africa because of a lack of surveillance infrastructure but also because of shared symptoms with other circulating pathogens (Plasmodium, dengue virus, CHIKV) that make differential diagnosis challenging. In such context, serological surveys, which quantify the level of antibodies in a population, can help evaluate levels of viral circulation and population immunity. Interpretation of these serosurveys is however impaired by important cross-reactivity between co-circulating viruses.

Here, we develop a statistical framework to characterize the circulation of ONNV in Mali in a context of high cross-reactivity with CHIKV, another alphavirus with high antigenic similarities[5]. This is done from the analysis of antibody titers to both ONNV and CHIKV with virus neutralization tests (VNT). Our statistical model can reconstruct the history of circulation of both viruses while explicitly accounting for the antibody dynamics due to infection and cross-reactivity. Our antibody dynamics model is informed by VNT from Martinique, a French territory in the West Indies where ONNV circulation is very unlikely[5]. This analysis demonstrates how the use of serosurveys for multiple pathogens and from different epidemiological contexts may help characterize viral response upon infection and the extent of cross-reactivity.

## Results

**Serology of ONNV and CHIKV.** We conducted a serological survey in seven sampling sites in Mali ($N = 793$) (Fig. 1a and Supplementary Table 1) and tested the presence of anti-CHIKV and anti-ONNV IgG antibodies using seroneutralization assays (Fig. 1b). In the classical seroprevalence classification method, a sample is considered negative if the titer is <20 for both viruses. A sample is considered CHIKV positive if the titer is >20 and four-fold or greater than the ONNV titer. A sample is considered ONNV positive if its titer is >20 and two-fold or greater than the CHIKV titer[6,7]. In other non-negative cases, samples are considered equivocal (Supplementary Figure 1). With this classification, we found that 1.8% ($N = 14$) of the surveyed population were positive for CHIKV, 26.9% ($N = 213$) were positive for ONNV, 10.1% ($N = 80$) were equivocal and 61.3% ($N = 486$) were negative for both (Table 1).

However, the samples from Martinique ($N = 62$) showed the limitations of the classical classification. This dataset consists of paired VNT of selected samples that were IgG CHIKV positive. 58 (94%) had a >20 ONNV titer (Fig. 1b). Based on the classical classification, 28 (44%) samples were CHIKV positive, 24 (38%) were equivocal, and 10 (16%) were ONNV positive (Table 1). However, there is no circulation of ONNV in Martinique and it is therefore likely that all positive samples were CHIKV positive and ONNV negative, and that the strong ONNV response is only due to cross-reactivity.

**Characterization of the response upon CHIKV and ONNV infections.** To characterize the antibody titer response to CHIKV and ONNV infections, one would ideally rely on the longitudinal follow-up of antibody titers of individuals after they have been infected by one of the viruses[8]. However, in the absence of such data that are costly and difficult to collect, we hypothesize that we can estimate parameters that drive this response from transversal multi-pathogen serological surveys such as the one presented in Fig. 1a. While in a context of important cross-reactivity, transversal data may not allow to precisely determine when and by which virus an individual was infected, the analysis of aggregated population level patterns can help reconstruct underlying mechanisms. For example, the fact that all individuals with high CHIKV titers have positive ONNV titers but that the reverse is not true (Fig. 1a) strongly hints at an asymmetric cross-reactive response.

We therefore developed a mathematical model that describes how the observed antibody titers depend on infection and cross-reactivity. We specifically model the infection status of an

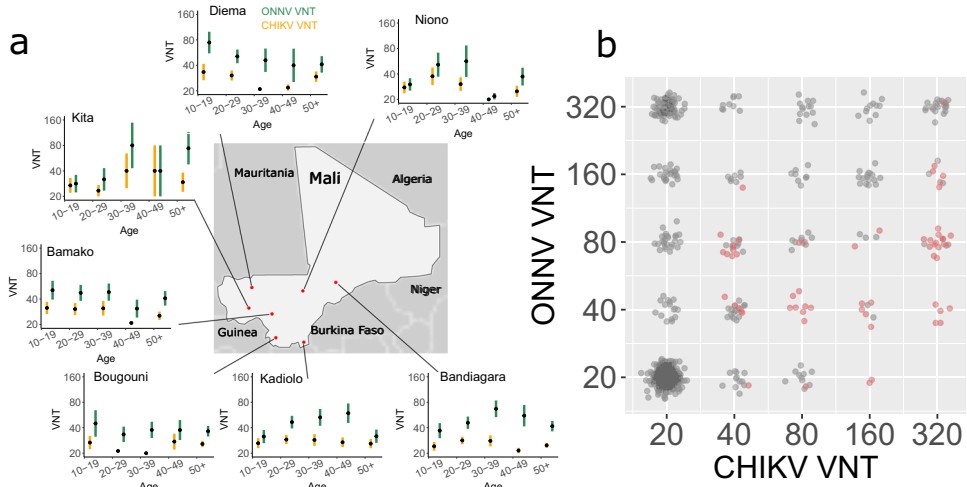

**Fig. 1 Serology of CHIKV and ONNV in Mali. a** Map of the seven sampling sites in Mali and virus neutralization titer (VNT, mean and standard error of the mean) of CHIKV (orange) and ONNV (green) by age in each site. Sample size for each age group and location is given in Supplementary Table 2. **b** Paired serology of CHIKV and ONNV in Mali (gray) and Martinique (red).

**Table 1 Comparison of the seroprevalence in Mali and Martinique using the classical method and model-based classification.**

| | Mali—Classical method | Mali—Model-based classification | Martinique—Classical method | Martinique—Model-based classification |
|---|---|---|---|---|
| **CHIKV Positive** | 1.8% (N = 14) | 13.3% (95% CrI: 9.4%–17.9%) | 45.2% (N = 28) | 98.4% (95% CrI: 94%–100%) |
| **CHIKV Negative** | 61.3% (N = 486) | 86.7% (95% CrI: 82.1%–86.7%) | 0% (N = 0) | 1.6% (95% CrI: 0%–5.7%) |
| **ONNV Positive** | 26.9% (N = 213) | 29.7 % (95% CrI: 25.3%–34.0%) | 16.1% (N = 10) | 0 (95% CrI: 0%–0%) |
| **ONNV Negative** | 61.3% (N = 486) | 70.3 % (95% CrI: 66.0%–74.7%) | 0% (N = 0) | 100% (95% CrI: 100%–100%) |
| **Equivocal** | 10.1% (N = 80) | — | 38.7% (N = 24) | — |

individual that can take one of four values: infected by CHIKV only, infected by ONNV only, infected by both CHIKV and ONNV, not infected (the model does not consider the timing of infections, only the final infection status). Key model parameters are how titers to both the infection and non-infecting virus change upon an historical infection, and also the prevalence of each virus in the population. The main model assumptions are summarized in Table 2. By comparing population level predictions for different sets of parameters to data (Fig. 1a), it is possible to estimate model parameters within a Bayesian Markov chain Monte Carlo (MCMC) inferential framework. In a simulation study, we showed that this approach successfully estimated model parameters from data (Supplementary Table 3).

We assumed that following infection, titers of the infecting virus increase above 20 according to a zero-truncated Poisson distribution on the log2 scale (Table 2). This model therefore assumes that the assays have a 100% sensibility and produce no false negatives. Titers above the observation threshold of 320 are set to 320. We assume that a cross-reactive response is observed in a fraction of the cases. In those cases where cross-reactivity happens, we assumed that titers for the non-infecting virus also increase following a zero-truncated Poisson distribution on the log2 scale, independent of the response against the infecting virus. We estimated that infection with ONNV increased the log2 ONNV titer by an average of 4.0 (95% CI: 3.6–4.3) and a standard deviation of 1.9 log2 titer (95% CrI: 1.8–2.0) (Fig. 2a). An infection with CHIKV also increased ONNV titer in 80% (95% CrI: 72%–87%) of the cases. The overall increase of ONNV titer due to a CHIKV infection was 1.6 log2 titer (95% CrI: 1.3–1.9) on average (Fig. 2d, f).

Infection with CHIKV increased CHIKV titers (increase of 2.4 log2 titer (95% CrI: 2.1–2.6) and standard deviation of 1.3 log2 titer (95% CrI: 1.2–1.4)) (Fig. 2e). An infection with ONNV increased CHIKV titer in 22% (95% CI: 8%–33%) of the cases, with an average increase of 0.61 log2 titer (95% CrI: 0.23–0.9) (Fig. 2b, c).

**Assessing virus circulation**. We used serocatalytic models to reconstruct the mode of circulation of the virus and retained a scenario where CHIKV and ONNV outbreaks occurred in recent years. In this model, all study participants were born at the time of the outbreak (since they are 15 y.o. or older) and the force of infection is independent of age. This is in agreement with the observation that for both viruses, the titers did not vary much with age (Fig. 3a). We estimated that 29.7 % (95% CrI: 25.3%–34.0%) of the surveyed population in Mali had been infected by ONNV (Fig. 3b). There was no major difference between the seroprevalence in the semi-arid regions in the North of Mali (33.3% (95% CrI: 27.6%–39.5%)) and in the tropical regions in the South (26.7% (95% CrI: 21.5%–31.9%)) (Odds Ratio OR: 1.34, 95% CrI: 0.99–1.73). No significant difference was observed between male and female participants (OR for females relative to males: 1.12; 95% CrI: 0.80–1.49).

With a prevalence of 13.3% (95% CrI: 9.4%–17.9%), CHIKV was less prevalent than ONNV. CHIKV was more likely to infect females (18.3% (95% CrI: 10.9%–26.2%)) than males (11.1% (95% CrI: 7.4%–15.4%)) (OR: 1.9, 95% CrI: 1.1–2.9) (Fig. 3a). The model was able to reproduce the observed titer distributions (Supplementary Fig. 2).

**Improving the interpretation of serological surveys**. Once parameters characterizing the antibody response, the cross-reactivity, and the force of infection were estimated, we explored if the model could be used to improve the interpretation of serological

**Table 2 Summary of the assumptions of the antibody response model and in the model of virus circulation. Alternative assumptions are tested in additional sensitivity analysis. VNT: Virus neutralization titer.**

| Submodel | Baseline assumptions | Alternative assumptions |
|---|---|---|
| Antibody response model | • Infection with a virus increases the VNT of the virus (direct response model) according to a zero-truncated Poisson distribution<br>• Infection with a virus increases the VNT of the other virus (cross-reactivity model) with a zero-truncated Poisson distribution<br>• Independence of the homologous and cross-reactive responses<br>• Only a fraction of infections lead to a cross-reactive response | • Different distributions of the response model (zero-truncated negative binomial)<br>• Cross-reactive response is proportional to the infecting virus antibody titer boost |
| Risk of infection | • No circulation of ONNV in Martinique<br>• No other virus with potential for cross-reactive response circulates<br>• CHIKV and ONNV outbreaks occurred in the recent years | • The annual probability of infection by CHIKV and ONNV is constant (model of endemic circulation)<br>• No CHIKV in Mali |

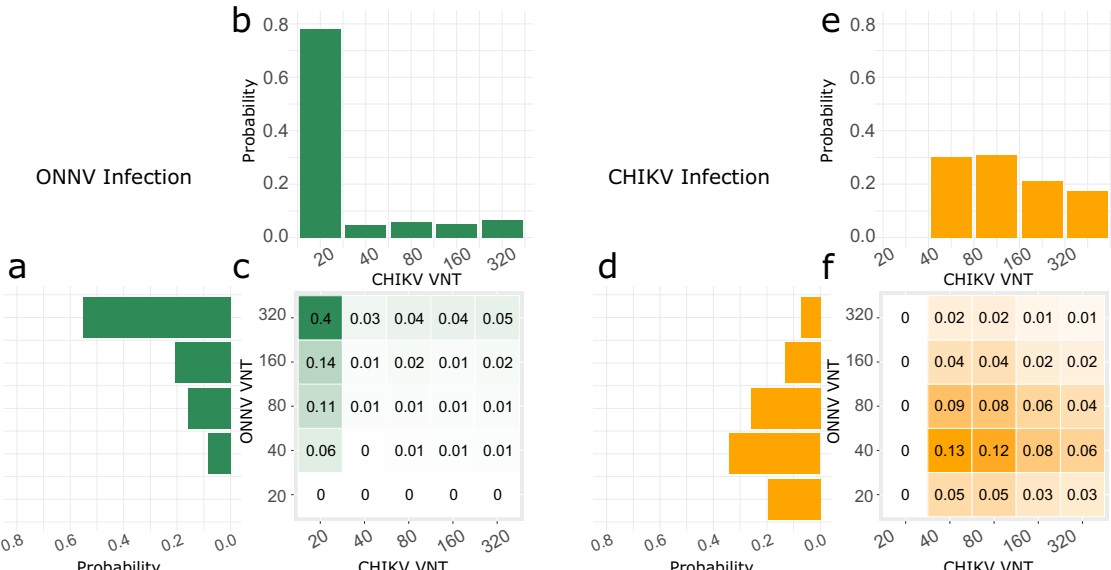

**Fig. 2 Response to infection.** The heatmaps represent the model estimates of the probability distribution of CHIKV and ONNV neutralization titers (VNT) following an ONNV (**a–c**, green) and a CHIKV infection (**d–f**, orange). They are the frequencies we would expect from samples obtained in a context where a single pathogen circulated. The histograms are the sum along rows and columns and display the probability distribution of ONNV response (**a**, **d**) and CHIKV response (**b**, **e**).

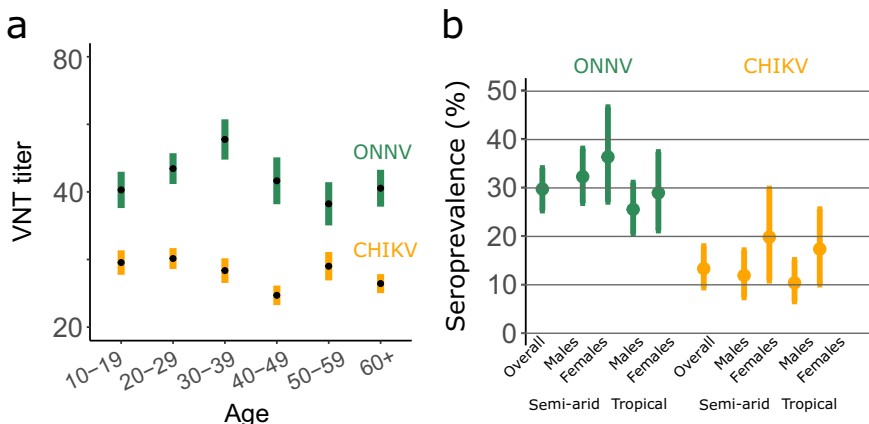

**Fig. 3 Model estimates of ONNV and CHIKV seroprevalence in Mali. a** Mean titer by age group (mean and standard error of the mean). Sample sizes for each age group are given in Supplementary Table 2. VNT: Virus neutralization titer. **b** Seroprevalence estimated with the model as a function of sex and living location. Dots represent the mean and error bars the 95% credible intervals.

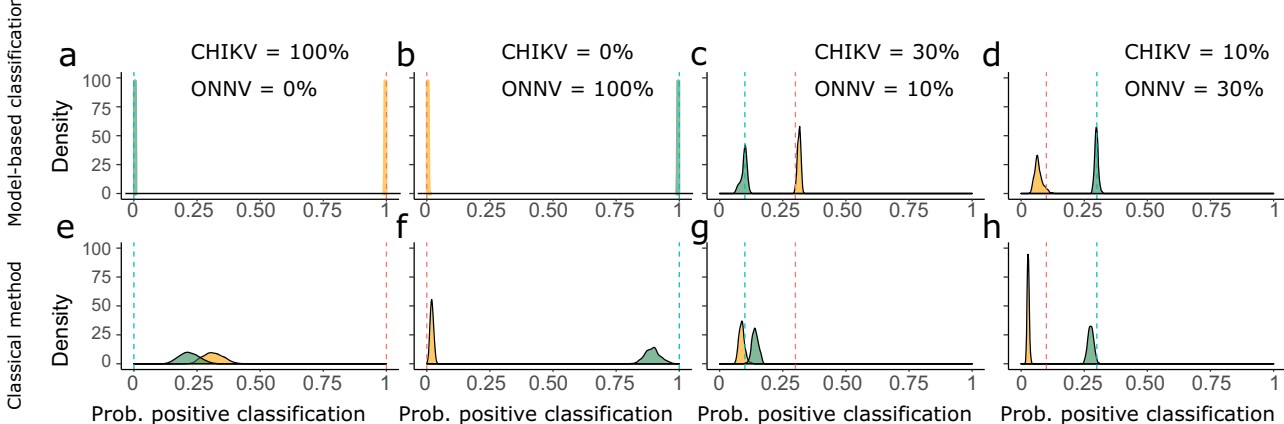

**Fig. 4 Performance of the classical and model-based classification methods in a simulation study.** The densities represent the posterior probability of classifying an individual as CHIKV (yellow) or ONNV (green) positive with the model (**a**–**d**) and the classical method (**e**–**h**) using simulated serological surveys. We considered four different scenarios where the prevalence of CHIKV and ONNV is (CHIKV: 100%; ONNV: 0%) in panels **a** and **e**, (CHIKV: 0%; ONNV: 100%) in panels **b** and **f**, (CHIKV: 30%; ONNV: 10%) in panels **c** and **g** and (CHIKV: 10%; ONNV: 30%) in panels **d** and **h**. The dashed vertical lines give the simulation values for the prevalence of CHIKV (yellow) and ONNV (green).

assays. Using the antibody model, we simulated serological surveys for various values of the prevalence. We developed a model-based classification of historical infection by CHIKV and/or ONNV that uses the estimated probability of being historically infected by ONNV or CHIKV for each value of the VNT (see Methods section Model-based classification). We classify individuals as positive or negative when their probability of infection is above 0.65 for CHIKV and 0.50 for ONNV, which were the thresholds that minimized the error for the estimated prevalence (Supplementary Fig. 3).

We then assessed the performance of our model-based classification of infection at estimating the prevalence in four scenarios of CHIKV and ONNV circulation (Fig. 4). In the first two scenarios where only CHIKV or ONNV circulates, the model-based classification perfectly retrieved the prevalence of the viruses (Fig. 4a, b). In contrast, when CHIKV prevalence was 100% (ONNV = 0%) the classical method recovered only 31% (95% CrI: 24%–40%) of CHIKV infections and misclassified 22% (95% CrI: 15%–30%) of them as ONNV positive (Fig. 4e). In the scenario of 100% ONNV prevalence and 0% CHIKV, the classical method was more accurate for the classification of ONNV infections (Fig. 4f): ONNV infections were correctly classified as ONNV positive in 90% (95% CrI: 85%–97%) of the cases and the probability that they were classified as CHIKV positive was only 2% (95% CrI: 0%–3%). In the scenarios of virus co-circulation, the model-based classification performed better than the classical method: For 30% CHIKV and 10% ONNV prevalence, the model estimated a CHIKV prevalence of 32% (95% CrI: 30%–33%), versus 9% (95% CrI: 7%–11%) with the classical model (Fig. 4c, g). It estimated an ONNV prevalence of 10% (95% CrI: 7%–12%) compared to 14% (12%–16%) for the classical method. For 10% CHIKV and 30% ONNV prevalence, we found a CHIKV prevalence of 7% (4%–11%) with the model, versus 3% (95% CrI: 2%–4%) with the classical method (Fig. 4d, h). ONNV prevalence was 30% (95% CrI: 28%–31%) with the model and 28% (95% CrI: 25%–30%) with the classical method.

**Sensitivity analysis**. Our results are in principle sensitive to specifications of the antibody response model that we explored in a series of sensitivity analyses. We first tested the hypothesis that anti-ONNV antibodies never induced a cross-reactive CHIKV response and showed that this alternative model is not well supported by the data (see Methods section Sensitivity analysis:

The probability of cross-reactivity). We also tested a model where the cross-reactive response was proportional to the infecting antibody titer boost of the infecting virus. The DIC of this model and the baseline model were similar, showing that these models explained the data equally well (see Methods section Sensitivity analysis: The cross-reactivity model). The interpretation of results in terms of prevalence and response parameters was very similar (Supplementary Table 4 and Supplementary Figure 4). Complementary studies would be needed to distinguish between these two models and understand the dynamics of the antibody response in detail. Third, we tested a model where the antibody response followed a zero-truncated negative distribution. We showed that the baseline model with a zero-truncated Poisson distribution had a better fit to the data (see Methods section Sensitivity analysis: The direct response model). Finally, we tested an additional model of endemic virus circulation. In this model, the annual probability of infection is constant and the exposure to the virus, and therefore the probability of an historical infection, increases with age. We showed that an age-independent model of infection offers a better fit to the data (Methods section Age-dependent force of infection).

## Discussion

Interpreting serological surveys is difficult due to the co-circulation of viruses and cross-reactivity in serological assays. In this study, we developed a statistical framework informed by data from regions with different patterns of virus circulation. This allowed us to estimate the prevalence of CHIKV and ONNV in Mali and quantify the impact of cross-reactivity on the observed response. Challenges associated with the cross-reactivity between CHIKV and ONNV have been noted in previous seroprevalence studies. For instance, Labeaud et al. used PRNT in a study in Kenya and found that 38% of samples were classified as equivocal. Here, we found that 10% of samples were equivocal for the Mali seroprevalence survey. In the dataset from Martinique, where samples were selected to be IgG CHIKV positive and where we do not expect the presence of ONNV, we found that 56% of the samples were classified as either ONNV positive or equivocal.

We estimated the probability of cross-reactivity and showed asymmetric cross-reactivity between CHIKV and ONNV consistent with previous studies. For example, in ref. [9], monoclonal antibodies prepared against ONNV and CHIKV epitopes were tested against both viruses using hemagglutination assays and

immunofluorescence. The authors showed that most antibodies prepared against CHIKV were able to neutralize ONNV, but only 54% of the antibodies prepared against ONNV recognized sites on CHIKV. Karabatsos[10] reported an asymmetry that was even larger, with the anti-CHIKV antibodies inhibition of ONNV being two orders of magnitude larger than that of CHIKV by anti-ONNV antibodies. We estimated in this study that 22% of ONNV infections gave rise to a CHIKV response.

We present here the first CHIKV seroprevalence study in the general population in Mali. We found high levels of CHIKV circulation in the country and a higher seroprevalence with our model than with the classical method (13.3% versus 1.8%, Table 1). Our results are consistent with a meta-analysis reporting a 16.4% seroprevalence in Africa (95% CI 9.1–25.2)[11]. To check that the higher seroprevalence estimate for CHIKV was not an artifact of the statistical method, we ran a simulation study assuming that CHIKV did not circulate and we were able to successfully estimate the prevalence (see Methods section CHIKV circulation in Mali). Moreover, we found that women had a higher risk of infection than men. Differential risks of CHIKV prevalence for men and women were observed across the world. There are situations where women are more frequently infected than men[12–15] others where men are more frequently infected than women[16], and situations were infection rates are similar in both genders[17]. The main explanation for differences lies in the conjunction of occupational and entomological factors. In places where *Aedes* mosquitoes are endophilic, women are usually more exposed to Chikungunya infection than men if they spend more time at home than men (e.g., if men are working in the fields)[6]. However, in places where Aedes mosquitoes are broadly exophilic (this is more frequently observed for *Aedes albopictus* in rural settings), men working in plantations may be more exposed than women staying at home[18]. The higher risk of infection in women is consistent with our limited knowledge of the entomological situation in Mali that points to exposure to endophilic Aedes aegypti mosquitoes.

We found high attack rates for ONNV while no report exists of ONNV infections. This highlights the need to reinforce surveillance and seroprevalence studies in this part of the world. It is however not surprising to find ONNV in Mali, a region with endemic malaria circulation and where Anopheles vectors are present[19]. ONNV titers did not vary with age. Such pattern could suggest that ONNV started circulating in the country only relatively recently, affecting all individuals irrespective of their age. However, since children were not included in the survey, this could have happened any time in the last 15 years and take the form of one large outbreak over that time period or multiple smaller outbreaks. In our approach we assumed that the titers did not decay with the time since infection. A scenario with a constant circulation of the virus and a loss of immunity would also result in a plateau in the age profile of seroprevalence, as can be observed for instance in malaria serosurveys[20]. Including this antibody decay could lead to different estimates of the attack rates and of the response parameters, but this remains speculative, as nothing is known about the long-term anti-ONNV antibody dynamics. To distinguish between these different scenarios more studies are needed, for instance by performing longitudinal studies or including children in a future survey.

Recently, modeling approaches have greatly increased our understanding of cross-reactive serological assays. For instance, these approaches helped improve diagnostic based on the value of serological assays[20], used population level data to inform individual diagnostic[21], informed risk of infection based on antibody dynamics[22–24], quantified the responses due to multiple influenza infections to detect recent infections[25]. Transversal serological studies can also provide insights on key epidemiological parameters, such as the attack rate and the mode of circulation of the viruses, by combining information from individuals from many locations and sociodemographic backgrounds[14,26]. Our approach shares many similarities with the general approach of Hay et al.[24], which proposes a statistical framework to study very diverse cross-reactive serological assays. However, the specificity of the viruses and assays used here required that we developed a unique antibody response and cross-reactivity model. In particular, we chose a zero-truncated Poisson model for the response and we specified in our model that cross-reactivity happens only for a fraction of the infections (approximately 80% of the CHIKV infections and 20% of the ONNV infections) (Table 2).

An accurate classification of past infections from a serological survey depends on the level of virus circulation in the region. As illustrated in the Martinique dataset, the classical method, which does not account for viral circulation, can classify a large proportion of actual CHIKV historical infections as ambiguous or ONNV positive. The tool we developed in this study explicitly assesses the prevalence and classifies individual titers accordingly. Arbovirus diagnosis is made difficult by the cross-reactivity with other viruses of the same family, and surveillance of a pathogen with a high risk of emergence is particularly challenging in regions of virus co-circulation[27]. In this context, our results highlight the importance of using serological assays of multiple cross-reactive pathogens from multiple locations.

## Methods

**Mali sera samples and ethics approval**. We conducted a cross sectional study in seven different eco-climatic localities in Mali from October to November 2016 to map emerging viruses circulation as previously reported in the case of Zika virus[28]. The same samples were used to assess CHIKV and ONNV seroprevalence.

We obtained the approval from The Institutional Review Board of the Faculty of Medicine and Odonto-Stomatology, University of Sciences, Techniques and Technologies, Bamako, Mali (IRB letter no. 2016/113/CE/FMPOS), and then visited all sites to explain the study context to health professionals, administrative authorities, and local community. Study areas and families were randomly selected in each district. Volunteers were recruited after obtaining community permission and provided signed informed consent after receiving information on the study in family language in the presence of a witness. We conducted the study according to institutional procedures and guidelines.

**Serological data**. Virus neutralization test (VNT): Of 793 Malian sera, 432 (45 doubtful and 387 positive, according to anti-CHIK IgG ELISA results) were selected to perform cytopathic effect (CPE) based VNT. Sensitivity and specificity values of this VNT were of 98.1% and 98.8%, respectively[29]. VNT for both viruses (CHIKV and ONNV) were performed using the same 432 sera. Diluted sera (1/10 to 1/160) was mixed with equal volume of 103 TCID50 viruses for 1 h; then 96 wells plates containing confluent Vero cells ATCC-CCL-81 were inoculated with viruses and serum mixture. CPE was read at day 5 post infection. Sera with a neutralizing titer ≥40 were considered positive. We performed also VNT for 62 sera (antibodies IgG CHIKV positive) from Martinique and Guadeloupe blood donors in the conditions.

**Statistical model of antibody response and measurement**. We model the serological response for the $N$ individuals in the survey. This model jointly estimates the probability of infection, the changes in antibody response induced by an infection, and accounts for the discrete structure of the data.

*Notations*. In our dataset, antibody titers take 5 values: 20, 40, 80, 160, and 320, the general formula being $20*2^n$ with $n = 0...4$. Antibody titers are labeled here with the exponent $n$. We consider an individual $j$. We note $n_j^C$ the value of the discrete CHIKV titer for individual $j$ ($n_j^O$ for ONNV). We write $\mathbf{X}_j$ a vector of characteristics of individual $j$. $\boldsymbol{\theta}$ is the vector parameter. We write $\mathbf{I}_j$ the infection status of individual $j$ which can take four possible values (0, 0), (1, 0), (0, 1) or (1, 1), where the first and second elements characterize infections by CHIKV and ONNV, respectively.

*Hierarchical model*. We build a 2-level Bayesian hierarchical model to characterize the joint distribution of antibody titers and the infection status of an individual.

We assume first that the infection status is known and write the model as

$$P\left(n_j^C, n_j^O, \mathbf{I}_j | \boldsymbol{\theta}, \mathbf{X}_j\right) = P\left(n_j^C, n_j^O | \mathbf{I}_j, \boldsymbol{\theta}, \mathbf{X}_j\right) P\left(\mathbf{I}_j | \boldsymbol{\theta}, \mathbf{X}_j\right). \quad (1)$$

The first submodel characterizes antibody response while the second describes the risk of infection.

*Antibody response model.* The antibody response model describes the increase in titers conditioned on the infection status.

1. In the absence of infection, we assume that both titers are equal to 0 and the probability of having titers $(n_j^C, n_j^O)$ is

$$P\left(n_j^C, n_j^O | \mathbf{I}_j = (0,0), \boldsymbol{\theta}, \mathbf{X}_j\right) = \begin{cases} 1 & \text{if } n_j^C = n_j^O = 0 \\ 0 & \text{otherwise} \end{cases} \quad (2)$$

2. When an individual is infected by CHIKV and not by ONNV, both CHIKV and ONNV titers can be boosted. The probability of the titers to be equal to $(n_j^C, n_j^O)$ is

$$P(n_j^C, n_j^O | \mathbf{I}_j = (1,0), \boldsymbol{\theta}, \mathbf{X}_j) = \text{Response}\,(n_j^C, \sigma^C) \times \left[p^{C \to O} \text{CR}(n_j^O, \sigma^{C \to O}) \right.$$
$$\left. + (1 - p^{C \to O}) \delta(n_j^O)\right], \quad (3)$$

where Response $(n_j^C, \sigma^C)$ is the probability for the direct response to a CHIKV infection to be equal to $n_j^C$. We account for the discrete structure of the data and model this response with a zero-truncated Poisson (ZTP) distribution, defined as a Poisson distribution conditioned on being nonzero:

$$\text{Response}(n_j^C, \sigma^C) = \text{Poisson}(n_j^C, \sigma^C | n_j^C > 0) \quad (4)$$

The truncation at zero translates the assumption that no false negatives can exist with the seroneutralisation assay. Cross-reactivity is modeled as an increase in ONNV titer in only a fraction $p^{C \to O}$ of CHIKV infected individuals, whereas in a fraction $1 - p^{C \to O}$ of CHIKV infected the ONNV titer does not increase. We write $\text{CR}(n_j^O, \sigma^{C \to O})$ the probability that the increase in ONNV titer due to cross-reactivity is equal to $n_j^O$, and assumed as for the direct response model a ZTP distribution which is determined by a single parameter $\sigma^{C \to O}$.

3. Similarly, if individual $j$ is infected by ONNV and not by CHIKV, the paired titer distribution is

$$P\left(n_j^C, n_j^O | \mathbf{I}_j = (0,1), \boldsymbol{\theta}, \mathbf{X}_j\right) = \text{Response}\,(n_j^O, \sigma^O)$$
$$\times \left[p^{O \to C} CR(n_j^C, \sigma^{O \to C}) + \left(1 - p^{O \to C}\right) \delta(n_j^C)\right], \quad (5)$$

where the response for ONNV depends on the parameter $\sigma^O$ that quantifies the increase in ONNV titer following ONNV infection. The cross-reactivity is modeled by a fraction $p^{O \to C}$ of ONNV infected that lead to a positive CHIKV response which is drawn with a ZTP of parameter $\sigma^{O \to C}$.

4. When an individual has been infected by both viruses, four scenarios can happen, depending on whether the assays are cross-reactive or not. In the case of cross-reactivity, the total response is the sum of the direct and the cross-reactive responses. We decompose these two responses by accounting for all possible values in the probability distribution which is therefore a convolution product given by

$$P\left(n_j^C, n_j^O | \mathbf{I}_j = (1,1), \boldsymbol{\theta}, \mathbf{X}_j\right) = (1 - p^{C \to O})(1 - p^{O \to C}) \text{Response}(n_j^C, \sigma^C)$$
$$\times \text{Response}(n_j^O, \sigma^O) + p^{C \to O}(1 - p^{O \to C}) \text{Response}(n_j^C, \sigma^C)$$
$$\times \sum_{k=0}^{n_j^O} \text{Response}(k, \sigma^O) \text{CR}(n_j^O - k, \sigma^{C \to O})$$
$$+ (1 - p^{C \to O}) p^{O \to C} \text{Response}(n_j^O, \sigma^O)$$
$$\times \sum_{k=0}^{n_j^C} \text{Response}(k, \sigma^C) \text{CR}(n_j^C - k, \sigma^{O \to C})$$
$$+ p^{C \to O} p^{O \to C} \sum_{k^C=0}^{n_j^C} \sum_{k^O=0}^{n_j^O} \text{Response}(k^C, \sigma^C) \text{Response}(k^O, \sigma^O)$$
$$\text{CR}(n_j^C - k^C, \sigma^{O \to C}) \text{CR}(n_j^O - k^O, \sigma^{C \to O}). \quad (6)$$

*Infection model.* The infection model describes the probability of infection by CHIKV and ONNV. We used serocatalytic models to reconstruct the annual force of infection defined as the per capita rate that a susceptible individual becomes infected. Including age in the infection model can provide insights on the mode of circulation of the virus. For instance, a constant exposure to the virus will lead to a gradual increase of seroprevalence with age. Alternatively, a punctual outbreak leads to a sudden increase in the seroprevalence. We considered those two models of virus circulation.

In the model with constant exposure, the force of infection $\lambda$ is constant with time. In that case, the probability of an historical infection of an individual is

expected to increase with their age

$$P(\text{age}) = 1 - e^{-\lambda \times \text{age}}. \quad (7)$$

In contrast, in the epidemic model, we assumed that CHIKV and ONNV outbreaks occurred in the recent years so that all individuals have been infected, independently of their age. If $\lambda$ is the force of infection at the time of the outbreak the probability of infection is then

$$P(\text{age}) = 1 - e^{-\lambda}. \quad (8)$$

Since we found that the probability of infection was independent of age (Fig. 3a), we relied on the epidemic model in our baseline analysis.

The force of infection additionally depends on the sociodemographic characteristics of individual $j$ $\mathbf{X}_j = (\text{location}_j, \text{sex}_j)$. We considered that risks were different between the semi-arid and tropical regions and between males and females. We note $f_{\text{sex}}^C(\text{females})$ the relative risk of CHIKV infection of females ($f_{\text{sex}}^C(\text{males}) = 1$), $f_{\text{region}}^C(\text{tropical})$ the relative risk of CHIKV infection for inhabitants of the tropical regions ($f_{\text{region}}^C(\text{semi} - \text{arid}) = 1$) and use the notation $f_{\text{sex}}^O$ and $f_{\text{region}}^O$ for the relative risks of ONNV infection.

We denote as $\Lambda_j^C$ the cumulative force of CHIKV infection for individual $j$ and $\Lambda_j^O$ the cumulative force of ONNV infection. In the model of constant circulation the cumulative force of infection is

$$\Lambda_j^C = \lambda^C \times f_{\text{sex}}^C(\text{sex}_j) \times f_{\text{region}}^C(\text{region}_j) \times \text{age}_j \quad (9)$$

and

$$\Lambda_j^O = \lambda^O \times f_{\text{sex}}^O(\text{sex}_j) \times f_{\text{region}}^O(\text{region}_j) \times \text{age}_j \quad (10)$$

The relation between the force of infection and the infection status is given by the set of four equations:

$$P(I_j = (0,0) | \boldsymbol{\theta}, \mathbf{X}_j) = e^{-\Lambda_j^C} e^{-\Lambda_j^O}$$
$$P(I_j = (1,0) | \boldsymbol{\theta}, \mathbf{X}_j) = (1 - e^{-\Lambda_j^C}) e^{-\Lambda_j^O}$$
$$P(I_j = (0,1) | \boldsymbol{\theta}, \mathbf{X}_j) = e^{-\Lambda_j^C} (1 - e^{-\Lambda_j^O}) \quad (11)$$
$$P(I_j = (1,1) | \boldsymbol{\theta}, \mathbf{X}_j) = (1 - e^{-\Lambda_j^C})(1 - e^{-\Lambda_j^O}).$$

*Likelihood.* To compute the likelihood we accounted for the fact that the infection status and the higher titer values are unobserved.

In practice, titers above 320 (4 in log scale) are set to 320. We denote $n_j^{C,\text{Obs}}$ the value of the observed CHIKV titer for individual $j$ ($n_j^{O,\text{Obs}}$ for ONNV). We account for censoring in an observation model that states that the probability of observing titers $(n_j^{C,\text{Obs}}, n_j^{O,\text{Obs}})$ is obtained by summing over the probability of the unobserved titers:

$$\text{P}\left(n_j^{C,\text{Obs}}, n_j^{O,\text{Obs}} | \mathbf{I}_j, \boldsymbol{\theta}, \mathbf{X}_j\right) = \sum_{n_j^C=0}^{+\infty} \sum_{n_j^O=0}^{+\infty} P\left(n_j^{C,\text{Obs}}, n_j^{O,\text{Obs}} | n_j^C, n_j^O\right) P\left(n_j^C, n_j^O | \mathbf{I}_j, \boldsymbol{\theta}, \mathbf{X}_j\right), \quad (12)$$

where the observation model is

$$P\left(n_j^{C,\text{Obs}}, n_j^{O,\text{Obs}} \middle| n_j^C, n_j^O\right) = P\left(n_j^{C,\text{Obs}} \middle| n_j^C\right) P\left(n_j^{O,\text{Obs}} \middle| n_j^O\right), \quad (13)$$

with $P(n_j^{C,\text{Obs}} | n_j^C) = \delta(n_j^{C,\text{Obs}} - n_j^C)$ if $n_j^C < 4$ and $P(n_j^{C,\text{Obs}} | n_j^C) = \delta(n_j^{C,\text{Obs}} - 4)$ otherwise. Similarly $P(n_j^{O,\text{Obs}} | n_j^O) = \delta(n_j^{O,\text{Obs}} - n_j^O)$ if $n_j^O < 4$ and $P(n_j^{O,\text{Obs}} | n_j^O) = \delta(n_j^{O,\text{Obs}} - 4)$ otherwise.

The contribution to the likelihood of individual $j$ is given by summing over the probability of the unobserved infection status and is therefore

$$\text{P}(n_j^{C,\text{Obs}}, n_j^{O,\text{Obs}} | \boldsymbol{\theta}, \mathbf{X}_j) = \sum_{\mathbf{I}_j} \sum_{n_j^C=0}^{+\infty} \sum_{n_j^O=0}^{+\infty} P(n_j^{C,\text{Obs}}, n_j^{O,\text{Obs}} | n_j^C, n_j^O) P(n_j^C, n_j^O | \mathbf{I}_j, \boldsymbol{\theta}, \mathbf{X}_j) P(\mathbf{I}_j | \boldsymbol{\theta}, \mathbf{X}_j).$$
$$\text{P}(n_j^{C,\text{Obs}}, n_j^{O,\text{Obs}} | \boldsymbol{\theta}, \mathbf{X}_j) = \sum_{\mathbf{I}_j} \sum_{n_j^C=0}^{+\infty} \sum_{n_j^O=0}^{+\infty} P(n_j^{C,\text{Obs}}, n_j^{O,\text{Obs}} | n_j^C, n_j^O) P(n_j^C, n_j^O | \mathbf{I}_j, \boldsymbol{\theta}, \mathbf{X}_j) P(\mathbf{I}_j | \boldsymbol{\theta}, \mathbf{X}_j).$$
$$(14)$$

In practice we sum up to $n_j^C, n_j^O = 15$.

**Parameter estimation.** We fitted the model parameters using a Markov chain Monte-Carlo (MCMC) framework implemented in the rstan package[30]. A No U-Turn sampler variant of Hamiltonian Monte-Carlo was used to update the parameters. Four chains of 20,000 iterations with a burnin of 10,000 iterations were ran. Prior distributions for the multiplicative factors for the cumulative force of infection ($f^C(\text{females}), f^O(\text{females}), f^C(\text{tropical}), f^O(\text{tropical})$) are lognormal with mean 0 and standard deviation 3. Flat priors were used for all other parameters ($\sigma^C, \sigma^O, \lambda^C, \lambda^O, p^{C \to O}, p^{O \to C}, \sigma^{C \to O}, \sigma^{O \to C}$).

**Model-based classification.** We used the model to establish a classification of infections based on the VNT of both viruses and on the prevalence. First, using the posterior distributions of the model parameters, we estimate the probability that a given individual has been infected by CHIKV or ONNV given the

titers($n_j^{C,\text{Obs}}, n_j^{O,\text{Obs}}$) are observed:

$$P(I|n_j^{C,\text{Obs}}, n_j^{O,\text{Obs}}, \boldsymbol{\theta}, \mathbf{X}_j) = \frac{P\left(n_j^{C,\text{Obs}}, n_j^{O,\text{Obs}}|I, \boldsymbol{\theta}, \mathbf{X}_j\right)P(I|\boldsymbol{\theta}, \mathbf{X}_j)}{Z}$$

$$= \frac{\sum_{n_j^C=0}^{+\infty}\sum_{n_j^O=0}^{+\infty} P\left(n_j^{C,\text{Obs}}, n_j^{O,\text{Obs}}|n_j^C, n_j^O\right)P\left(n_j^C, n_j^O|I, \boldsymbol{\theta}, \mathbf{X}_j\right)P(I|\boldsymbol{\theta}, \mathbf{X}_j)}{Z}, \quad (15)$$

where the denominator $Z = \sum_k P(n_j^{C,\text{Obs}}, n_j^{O,\text{Obs}}|\mathbf{I}_k, \boldsymbol{\theta}, \mathbf{X}_j)P(\mathbf{I}_k|\boldsymbol{\theta}, \mathbf{X}_j)$ is the sum over the four infection statuses and the probability of observing titers is given in Eqs. (2,4,5,6) above.

We simulated surveys for various levels of prevalence and estimated the probability of infection for each pair of VNT. We classified individuals as positive or negative depending on the value relative to a threshold and compared this classification to the real, simulated historical infections. The optimal threshold was defined as the one that minimized the difference between the estimated prevalence and the input prevalence. We used a threshold of 0.65 for CHIKV probability of infection (Supplementary Fig. 3a) as it gives a good estimation for realistic values of CHIKV prevalence (Supplementary Fig. 3b for a prevalence below 30%).

The threshold for the probability of an historical ONNV infection also depends on the prevalence (Supplementary Fig. 3c). We define in our model-based classification an individual as ONNV positive if the probability of ONNV infection is above 0.5 and found again a good agreement between input and estimated prevalence (Supplementary Fig. 3d).

**Model comparison and sensitivity analysis**. To ensure the robustness of our results to modeling assumptions, we explored and compared different model variants in a sensitivity analysis. The four model variants are documented below and their DICs compared in Supplementary Table 5:

i.   The direct response model
     We tested the robustness of our estimates by setting a different distribution function for the antibody response following an infection. We ran a similar analysis for the response model using a zero-truncated negative binomial distribution (ZTNB), defined as

$$\text{ZTNB}(n \,|\, p, \psi) = \frac{1}{1 - p^\psi}\text{Negative Binomial}(n \,|\, p, \psi) \text{ if } n > 0$$

     The value of the DIC shows a stronger support for the baseline, ZTP model (DIC = 3099 vs 3092 for the baseline model).

ii.  The probability of cross-reactivity
     We assumed in this model a one-way cross-reactivity in which CHIKV infection can induce a response in ONNV titer, and the probability $p^{O \to C}$ of an ONNV infection to induce a CHIKV response is set to 0. The value of the DIC shows a weaker support for the model with one-way than two-way cross-reactivity (DIC = 3098 vs 3092 for the baseline model). Most parameters were similar between both models except for CHIKV prevalence in Mali that was estimated to be higher in the case of one-way cross-reactivity (18.9% (95% CrI: 16.3%–21.6%) vs 13.3% (95% CrI: 9.4%–17.9%) for the baseline model, see Supplementary Table 4).

iii. The cross-reactivity model
     In the baseline model, we assumed that the cross-reactive response was independent of the response against the infecting virus. We tested an additional model where the cross-reactive response was a ZTP with a parameter proportional to the observed response for the infecting virus. For instance, in the case of an infection by CHIKV, the probability density function of the cross-reactive ONNV response is

$$\text{CR}(n_j^O, n_j^C \cdot \sigma^{C \to O}) = \text{Poisson}\left(n_j^O, \sigma^{C \to O} \cdot n_j^C + \varepsilon|n_j^C > 0\right), \quad (16)$$

     where $\varepsilon = 0.01$ is a small term that was added to ensure that the parameter of the Poisson distribution is non-zero. The probability density function for the VNT of the two viruses is

$$P(n_j^C, n_j^O|\mathbf{I}_j = (1, 0), \boldsymbol{\theta}, \mathbf{X}_j) = \text{Response}(n_j^C, \sigma^C) \times \left[p^{C \to O}\text{CR}(n_j^O, n_j^C \cdot \sigma^{C \to O})\right.$$
$$\left. + (1 - p^{C \to O})\delta(n_j^O)\right]. \quad (17)$$

     The value of the DIC does not show a stronger support for this model than for the baseline model (DIC = 3089 versus 3092 for the baseline model).

iv.  Age-dependent force of infection

We tested two models of the annual variations of the force of infection. First a model of constant exposure to the virus, and second, an age-independent model. We found no evidence of a constant exposure to the virus (DIC = 3231 vs 3092 for the age-independent model, considered here as the baseline model).

**Evaluation of the statistical framework**. We performed a simulation study to evaluate our statistical framework. We simulated 100 surveys in a population with the same characteristics (sex, location) as in the dataset and with parameters equal

to the median of the posterior distribution. For each simulated dataset, we estimated the median of the posterior distribution of each parameter and assessed the ability of the model to retrieve the input parameters. Parameters were well estimated (Supplementary Table 3), with input parameters falling into the 95% credible intervals at least in 88% of the simulations.

**Model adequacy**. We assessed model adequacy by simulating 2,000 samples with parameters drawn from the posterior distributions. We compared the observed number of samples with given CHIKV and ONNV titer values with the simulations (Supplementary Table 6). In most cells (22/25) the observed values are in the predicted range.

**CHIKV circulation in Mali**. To assess whether the estimated CHIKV prevalence in Mali was not an artifact of the model, we tested a scenario where only ONNV circulated in Mali and only CHIKV circulated in Martinique. Sample titers were simulated using the median of the posterior distributions for the parameters of the response. N = 856 samples were simulated (with N = 793 for Mali and N = 62 for Martinique). We found that the higher density interval of the posterior distribution of Mali CHIKV prevalence contains 0 (mean = 0.01, 95% CrI 0–0.05) and concluded that the model correctly estimated that CHIKV did not circulate in Mali (see Supplementary Fig. 5).

**Reporting summary**. Further information on research design is available in the Nature Research Reporting Summary linked to this article.

## Data availability
A dataset is available in a format that maintains anonymity of survey participants from the GitHub link at https://github.com/nathoze/ONNV_CHIKV. For each individual the dataset contains: age group (10-year classes), ONNV VNT, CHIKV VNT, region (North or South), sex (Code and data available at https://doi.org/10.5281/zenodo.5356320).

## Code availability
Code is available from the GitHub link at https://github.com/nathoze/ONNV_CHIKV. All models were implemented in R version 3.6.1 (Code and data available at https://doi.org/10.5281/zenodo.5356320).

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

## Acknowledgements

We dedicate this article to Ogobara K. Doumbo, who initiated this project before he passed. May he rest in peace. We thank Christine Isnard from EFS, Marseille for invaluable technical contribution. We are grateful to Ismaila Thera, from MRTC, Bamako who developed the electronic database using ODK and provided the data management service for the study. We also thank the study district health officers and the study population for their cooperation. Specifically, we are indebted to Hamma Maiga, Bakary Sidibé, Modibo Diarra, Kassoum Kayentao, Souleymane Dama, Hamidou Niangaly, Amadou Bamadio, Hamadoun Diaité, Karim Traoré and Balla Diarra for their support to field investigations. Nathanaël Hozé and Simon Cauchemez acknowledge financial support from the AXA Research Fund, the Investissement d'Avenir program, the Laboratoire d'Excellence Integrative Biology of Emerging Infectious Diseases program (Grant ANR-10-LABX-62-IBEID), the INCEPTION project (PIA/ANR-16-CONV-0005), We thank the European Union's Horizon 2020 research and innovation program under ZIKAlliance grant agreement No. 734548 and VEO grant agreement No. 874735 and the Minister of Health and Hygiene of Mali supported this work through the subvention no. 2016/668116-0 from the Mali World Health Organization Local Office.

## Author contributions

N.H., X.L., S.C., I.D., A.K.S., B.P., L.P., I.S., A. Da., A. Dj., B.K., O.K.D. and M.A.T. performed research. N.H., X.L. S.C. and I.D. analyzed the data and wrote the paper. S.C. and X. L. are equally contributing senior authors.

## Competing interests

The authors declare no competing interests.
