## [Peer Review File · Nature Communications]

REVIEWER COMMENTS

Reviewer #1 (Remarks to the Author):

This paper by Hoze et al. describes an antibody dynamics model to address the challenges to viral disease surveillance of high degrees of cross-reactivity between closely related viruses in serology tests, in this case chikungunya and o'nyong-nyong viruses. The modeling focuses on two locations: Mali and Martinique, which the authors describe as endemic for chikungunya but without known o'nyong-nyong (but likely in Mali). These viruses are good choices in that there is strong serologic cross-reactivity, but past work has shown that their relationships can be asymmetric, with neutralizing antibodies generated against known infection by one virus sometimes of lower titer against the homologous virus than against the heterologous virus. Thus, the correct infection history cannot always be inferred even when plaque-reduction neutralization tests are performed against both viruses. The assumption that o'nyong-nyong does not occur in Martinique is probably correct, and both viruses are almost certainly present in Mali, at least occasionally, but information on endemic exposure of both is very limited. Thus, how can assumptions regarding the initially infecting and reinfecting viruses be made? In particular, the assumption that cross-reactive rises antibody neutralizing titers occur inconsistently and randomly is not borne out with experimental work, where cross-reactivity is very consistent. Therefore, it is difficult to understand how sera from single timepoints can reliably be used to determine changes in antibody titers with reinfections by either virus. It is also unclear if these two viruses, with their unusual asymmetric cross-reactivities, are the best to develop this model. Overall, the paper is well written and appropriately brief, although I believe that the model assumptions deserve much more discussion.

Minor comments:

1. Line 73: How was it determined that the neutralizing antibodies were IgG?
2. Line 101: The assumptions underlying the model should be clearly stated here.
3. Line 135: In known CHIKV-endemic locations this virus generally infects males at a higher rate; this should be discussed.

Reviewer #2 (Remarks to the Author):

This paper by Hozé et al. describes an interesting seroprevalence survey of an understudied virus in an understudied region and presents novel methods to deal with the unique asymmetrical cross reactivity that hampers the interpretation of many serological datasets from regions where similar viruses co-circulate. The analyses is robust, described in extensive detail and give interesting results. I have only one or two comments and clarifications to add to what is a good manuscript.

Major comments:

The Introduction (and/or discussion) is missing a section on existing statistical and model-based approaches to interpret cross-reactive serological data (to give just 2 of what I suspect is many: e.g. Hay et al. PLoS Comp Bio 2020 <https://journals.plos.org/ploscompbiol/article?id=10.1371/journal.pcbi.1007840>, Henderson et al. Nature communications 2021 <https://www.nature.com/articles/s41467-021-21788-y>). While the method described here is valid and clearly useful, there are alternative frameworks out there for these kinds of questions. A formal quantitative comparison is probably beyond the scope of this paper, but it would be useful to see a discussion of the strengths and weaknesses of different existing approaches.

Line 108: "independent of the response against the infecting virus "- what is the evidence for or against this assumption that cross reactivity is independent of infecting virus antibody titre boost? Intuitively one might expect more robust generalised responses to be more cross reactive.

Minor comments:

Add equivocal results to table 1?- surely part of the benefit of the model-based classification is the reduced equivocal classification? It would also be helpful to make it clear what the total number of sera per site are in this table.

Figure 4 legend- "E-F" -> "E-H"

Line 267: "the classical method was also more accurate at classifying ONNV infections" – do you mean model based method here- both look good, but model slightly better?

No risk factor analysis for Martinique (i.e. equivalent of fig 3)?

Check equation 5 and 6- $p \rightarrow \sigma$?

REVIEWER COMMENTS

Reviewer #1 (Remarks to the Author):

This paper by Hoze et al. describes an antibody dynamics model to address the challenges to viral disease surveillance of high degrees of cross-reactivity between closely related viruses in serology tests, in this case chikungunya and o'nyong-nyong viruses. The modeling focuses on two locations: Mali and Martinique, which the authors describe as endemic for chikungunya but without known o'nyong-nyong (but likely in Mali). These viruses are good choices in that there is strong serologic cross-reactivity, but past work has shown that their relationships can be asymmetric, with neutralizing antibodies generated against known infection by one virus sometimes of lower titer against the homologous virus than against the heterologous virus. Thus, the correct infection history cannot always be inferred even when plaque-reduction neutralization tests are performed against both viruses.

The assumption that o'nyong-nyong does not occur in Martinique is probably correct, and both viruses are almost certainly present in Mali, at least occasionally, but information on endemic exposure of both is very limited.

Thus, how can assumptions regarding the initially infecting and reinfecting viruses be made?

We did not make any assumption in the model about the initially infecting and the reinfecting viruses. Rather we model the response for a participant who would have been infected by both viruses, by one of the viruses or by none of them. We do not make any more assumptions, for instance about the time when infections occurred or the order in which infections occurred.

These points have been clarified in the revised manuscript:

“We therefore developed a mathematical model that describes how the antibody titers of an individual depend on their infection status, that can take one of four values: infected by CHIKV only, infected by ONNV only, infected by both CHIKV and ONNV, not infected (the model does not consider the timing of infections, only the final infection status).”

In particular, the assumption that cross-reactive rises antibody neutralizing titers occur inconsistently and randomly is not borne out with experimental work, where cross-reactivity is very consistent.

We agree with the referee that the sentence “Cross-reactivity is a random event” lacked clarity. We therefore removed it from the manuscript.

The ONNV assays in the survey from Martinique (red dots in Fig 1B) can take many possible values: for instance, for a CHIKV titer of 40, ONNV titers are distributed between 20 and

160, although it is very likely that these titers are only due to cross-reactivity and not to a direct infection by ONNV. These differences in the seroneutralisation responses across individuals may be due to sex, age, time since infection, multiple prior exposures to the virus. Our probabilistic approach ensures that we can properly capture these individual variations.

Therefore, it is difficult to understand how sera from single timepoints can reliably be used to determine changes in antibody titers with reinfections by either virus.

Thank you for this important point. We agree that, in the original manuscript, we did not sufficiently explain how a single time point could be used to evaluate the cross reactivity viruses. In the revised manuscript, we added two paragraphs to explain how our approach allows to estimate key parameters of interest from transversal serological data:

“To characterize the antibody titer response to CHIKV and ONNV infections, one would ideally rely on the longitudinal follow-up of antibody titers of individuals after they have been infected by one of the viruses [8]. However, in the absence of such data that are costly and difficult to collect, we hypothesize that we can estimate parameters that drive this response from transversal multi-pathogen serological surveys such as the one presented in Figure 1A. While in a context of important cross-reactivity, transversal data may not allow to precisely determine when and by which virus an individual was infected, the analysis of aggregated population level patterns can help reconstruct underlying mechanisms. For example, the fact that all individuals with high CHIKV titers have positive ONNV titers but that the reverse is not true (Figure 1A) strongly hints at an asymmetric cross-reactive response.

We therefore developed a mathematical model that describes how the antibody titers of an individual depends on their infection status, that can take one of four values: infected by CHIKV only, infected by ONNV only, infected by both CHIKV and ONNV, not infected (the model does not consider the timing of infections, only the final infection status). Key model parameters include the prevalence of each virus in the population, and parameters describing how titers to both the infection and non-infecting virus change upon an historical infection. Key model assumptions are summarized in Table 1. By comparing population level predictions for different sets of parameters to data (Figure 1A), it is possible to estimate model parameters within a Bayesian Markov chain Monte Carlo (MCMC) inferential framework.”

In addition, we demonstrated in a simulation study that our approach could indeed estimate key parameters of interest from transversal serological data:

« In a simulation study, we showed that this approach successfully estimated key model parameters from data (Supplementary Table 2). »

Finally, a number of modelling studies have already shown how transversal serological data can provide insights on some of the key model parameters we are interested in here, supporting our approach. We added these references in the discussion:

Transversal serological studies can also provide insights on key epidemiological parameters, such as the attack rate and the mode of circulation of the viruses, by combining information from individuals from many locations and sociodemographic backgrounds [14,15].

It is also unclear if these two viruses, with their unusual asymmetric cross-reactivities, are the best to develop this model.

The aim of this paper is to better understand the complex patterns of cross-reactivity between CHIKV and ONNV. The unusual asymmetric cross-reactivity between the two viruses makes the analysis particularly interesting and challenging and requires the development of a dedicated model that fully capture the specific features of ONNV and CHIKV. We present such a framework in this manuscript.

The unusual asymmetric cross-reactivity makes it even more important to have an analytical framework that can correctly account for this phenomenon. Without such corrections, key estimates would be biased. For example, the classical method concludes that there is important circulation of ONNV in Martinique and little circulation of CHIKV in Mali. Our framework corrects for these problems, as summarized in our abstract:

“Given cross-reactivity, naïve analysis of the data would wrongly conclude important circulation of ONNV in Martinique and low prevalence (1.8%) of CHIKV in Mali. In contrast, our model can explain high ONNV titers in Martinique even though the virus is absent from the island and estimates that, in Mali, the prevalence of ONNV and CHIKV is 29.7 % (95% credible interval (CrI): 25.3% - 34.0%) and 13.3% (95% CrI: 9.4% - 17.9%), respectively. In Mali, females are 1.9 (95% CrI: 1.1 – 2.9) times more likely to be infected by CHIKV than males. While a CHIKV infection induces an ONNV response in 80% of cases, an ONNV infection leads to a cross-reactive CHIKV response in only 22% of cases.”

Overall, the paper is well written and appropriately brief, although I believe that the model assumptions deserve much more discussion.

Thank you. We agree that the different assumptions lacked clarity. We add two paragraphs to clarify our approach:

“To characterize the antibody titer response to CHIKV and ONNV infections, one would ideally rely on the longitudinal follow-up of antibody titers of individuals after they have been infected by one of the viruses [8]. However, in the absence of such data that are costly and difficult to collect, we hypothesize that we can estimate parameters that drive this response from transversal multi-pathogen serological surveys such as the one presented in Figure 1A. While in a context of important cross-reactivity, transversal data may not allow to precisely determine when and by which virus an individual was infected, the analysis of aggregated population level patterns can help reconstruct underlying mechanisms. For example, the fact

that all individuals with high CHIKV titers have positive ONNV titers but that the reverse is not true (Figure 1A) strongly hints at an asymmetric cross-reactive response.

We therefore developed a mathematical model that describes how the antibody titers of an individual depends on their infection status, that can take one of four values: infected by CHIKV only, infected by ONNV only, infected by both CHIKV and ONNV, not infected (the model does not consider the timing of infections, only the final infection status). Key model parameters include the prevalence of each virus in the population, and parameters describing how titers to both the infection and non-infecting virus change upon an historical infection. Key model assumptions are summarized in Table 1. By comparing population level predictions for different set of parameters to data (Figure 1A), it is possible to estimate model parameters within a Bayesian Markov chain Monte Carlo (MCMC) inferential framework.”

We also added a table summarizing the main model assumptions as well as a list of alternative assumptions we tested in sensitivity analyses. The table is

Table 1. Summary of the assumptions of the antibody response model and in the model of virus circulation. Alternative assumptions are tested in additional sensitivity analysis.

Submodel	Baseline assumptions	Alternative assumptions
Antibody response model	 • Infection with a virus increases the VNT of the virus (direct response model) according to a zero-truncated Poisson distribution • Infection with a virus increases the VNT of the other virus (cross-reactivity model) with a zero-truncated Poisson distribution) • Independence of the homologous and cross-reactive responses • Only a fraction of infections lead 	 • Different distributions of the response model (negative binomial) • Cross-reactive response is proportional to the infecting virus antibody titer boost

	to a cross-reactive response	
Risk of infection	 • No circulation of ONNV in Martinique • No other virus with potential for cross-reactive response circulates • CHIKV and ONNV outbreaks occurred in the recent years 	 • The annual probability of infection by CHIKV and ONNV is constant (model of endemic circulation) • No CHIKV in Mali

These assumptions are described in the results section:

We assumed that following infection, titers of the infecting virus increase above 20 according to a zero-truncated Poisson distribution on the log2 scale (Table 1). To model cross-reactivity, we assumed that titers for the non-infecting virus increase following a zero-truncated Poisson distribution on the log2 scale, independent of the response against the infecting virus.

We extended the discussion on the model assumptions – We now write in the discussion :

In our approach we assumed that the titers did not decay with the time since infection. A scenario with a constant circulation of the virus and a loss of immunity would also result in a plateau in the age profile of seroprevalence, as can be observed for instance in malaria serosurveys [20]. Including this antibody decay could lead to different estimates of the attack rates and of the response parameters.

In the sensitivity analysis, we tested different assumptions of the antibody response model

Third, we tested a model where the antibody response followed a zero-truncated negative distribution. We showed that the baseline model with a zero-truncated Poisson distribution had a better fit to the data (see Methods section Sensitivity analysis: The direct response model)

We also added sensitivity analysis where we discuss the assumptions on the cross-reactive antibody response model:

Our results are in principle sensitive to specifications of the antibody response model that we explored in a series of sensitivity analyses. We first tested the hypothesis that anti-ONNV antibodies never induced a cross-reactive CHIKV response and showed that this alternative model is not well supported by the data (see Methods section Sensitivity analysis: The probability of cross-reactivity). We also tested a model where the cross-reactive response was proportional to the infecting antibody titer boost of the infecting virus. The DIC of this model and the baseline model were similar, showing that these models explained the data equally well (see Methods section Sensitivity analysis: The cross-reactivity model). The interpretation of results in terms of prevalence and response parameters was very similar (Supplementary Table 3 and Supplementary Figure 4). Complementary studies would be needed to distinguish between these two models and understand the dynamics of the antibody response in detail.

Minor comments:

1. Line 73: How was it determined that the neutralizing antibodies were IgG?

In the course of Chikungunya infection "Neutralizing IgM starts to appear as early as day 4 of symptoms, and their appearance from day 6 is associated with a reduction in viremia. IgM acts in a complementary manner with the early IgG, but plays the main neutralizing role up to a point between days 4 and 10, which varies between individuals. After this point, total neutralizing capacity is attributable almost entirely to the robust neutralizing IgG response" (Chong-Long Chua, I-Ching Sam, Chun-Wei Chiam, Yoke-Fun Chan. PLOS ONE ; DOI:10.1371/journal.pone.0171989).

In our study, we included only healthy adults (*i.e.* no fever, no acute disease). According to Chua *et al.* and to what is usually observed in the immune response that follows acute viral infections, it is therefore expected that the observed seroneutralizing activity is mostly due to specific IgG antibodies.

2. Line 101: The assumptions underlying the model should be clearly stated here.

Thank you for the suggestion. We agree that the section explaining the assumptions lacked clarity. We extended this section and now write:

To characterize the antibody titer response to CHIKV and ONNV infections, one would ideally rely on the longitudinal follow-up of antibody titers of individuals after they have been infected by one of the viruses [10]. However, in the absence of such data that are costly and difficult to collect, we hypothesize that we can estimate parameters that drive this response from transversal multi-pathogen serological surveys such as the one presented in Figure 1A. While in a context of important cross-reactivity, transversal data may not allow to precisely determine when and by which virus an individual was infected, the analysis of aggregated

population level patterns can help reconstruct underlying mechanisms. For example, the fact that all individuals with high CHIKV titers have positive ONNV titers but that the reverse is not true (Figure 1A) strongly hints at an asymmetric cross-reactive response.

We therefore developed a mathematical model that describes how the antibody titers of an individual depend on their infection status, that can take one of four values: infected by CHIKV only, infected by ONNV only, infected by both CHIKV and ONNV, not infected (the model does not consider the timing of infections, only the final infection status). Key model parameters include the prevalence of each virus in the population, and parameters describing how titers to both the infection and non-infecting virus change upon an historical infection. Key model assumptions are summarized in Table 1. By comparing population level predictions for different set of parameters to data (Figure 1A), it is possible to estimate model parameters within a Bayesian Markov chain Monte Carlo (MCMC) inferential framework. In a simulation study, we showed that this approach successfully estimated key model parameters from data (Supplementary Table 12). We assumed that following infection, titers of the infecting virus increase above 20 according to a zero-truncated Poisson distribution on the log₂ scale (Table 1). To model cross-reactivity, we assumed that titers for the non-infecting virus increase following a zero-truncated Poisson distribution on the log₂ scale, independent of the response against the infecting virus.

We also added a table explaining the different assumptions of the model and the alternative assumptions that were tested in sensitivity analysis.

Table 1. Summary of the assumptions in the antibody response model and in the model of virus circulation. Alternative assumptions are tested in additional sensitivity analysis.

Submodel	Baseline assumptions	Alternative assumptions
Antibody response model	 • Infection with a virus increases the VNT of the virus (direct response model) according to a zero-truncated Poisson distribution • Infection with a virus increases the VNT of the other virus (cross-reactivity model) with a zero-truncated Poisson distribution) 	 • Different distributions of the response model (negative binomial) • Cross-reactive response is proportional to the infecting virus antibody titer boost

	 • Independence of the homologous and cross-reactive responses • Only a fraction of infections lead to a cross-reactive response 	
Risk of infection	 • No circulation of ONNV in Martinique • No other virus with potential for cross-reactive response circulates • CHIKV and ONNV outbreaks occurred in the recent years 	 • The annual probability of infection by CHIKV and ONNV is constant (model of endemic circulation) • No CHIKV in Mali

3. Line 135: In known CHIKV-endemic locations this virus generally infects males at a higher rate; this should be discussed.

A review of the literature reveals contrasting findings regarding the seroprevalence of Chikungunya in men and women. There are situations where women are more frequently infected than men (*Khatun et al., PLOS NTD, 2015; Sergon et al., The American journal of tropical medicine and hygiene, 2007*), others where men are more frequently infected than women (*Asebe et al, PLOS One, 2021*), and situations where infection rates are similar in both genders (*Sergon et al., The American journal of tropical medicine and hygiene, 2008*).

The main explanation for differences lies in the conjunction of occupational and entomological factors. In places where *Aedes* mosquitoes are endophilic, women are usually more exposed to Chikungunya infection than men if they spend more time at home than men (e.g., if men are working in the fields). However, in places where *Aedes* mosquitoes are broadly exophilic (this is more frequently observed for *Ae albopictus* in rural settings), men working in plantations may be more exposed than women staying at home. Many authors have reported convergent observations in this line (*A. Desiree LaBeaud et al., PLoS Negl Trop Dis., 2015 ; doi:10.1371/journal.pntd.0003436. Idris Nasir Abdullahi et al; doi: 10.1080/20477724.2020.1743087.*)

Our results (*i.e.*, seroprevalence is higher in women than men) suggest that exposure to endophilic mosquitoes would constitute the primary risk factor for Chikungunya infection in Mali. This is coherent with our limited knowledge of the entomological situation that points to exposure to endophilic *Aedes aegypti* mosquitoes.

We now write in the paper

Differential risks of CHIKV prevalence for men and women were observed across the world. There are situations where women are more frequently infected than men [12,13, 14, 15] others where men are more frequently infected than women [16], and situations where infection rates are similar in both genders [17]. The main explanation for differences lies in the conjunction of occupational and entomological factors. In places where Aedes mosquitoes are endophilic, women are usually more exposed to Chikungunya infection than men if they spend more time at home than men (e.g., if men are working in the fields) [6]. However, in places where Aedes mosquitoes are broadly exophilic (this is more frequently observed for Aedes albopictus in rural settings), men working in plantations may be more exposed than women staying at home [18]. The higher risk of infection in women is consistent with our limited knowledge of the entomological situation in Mali that points to exposure to endophilic Aedes aegypti mosquitoes.

Reviewer #2 (Remarks to the Author):

This paper by Hozé et al. describes an interesting seroprevalence survey of an understudied virus in an understudied region and presents novel methods to deal with the unique asymmetrical cross reactivity that hampers the interpretation of many serological datasets from regions where similar viruses co-circulate. The analysis is robust, described in extensive detail and gives interesting results. I have only one or two comments and clarifications to add to what is a good manuscript.

We thank the reviewer for these encouraging comments.

Major comments:

The Introduction (and/or discussion) is missing a section on existing statistical and model-based approaches to interpret cross-reactive serological data (to give just 2 of what I suspect is many: e.g. Hay et al. PLoS Comp Bio 2020 <https://journals.plos.org/ploscompbiol/article?id=10.1371/journal.pcbi.1007840>, Henderson et al. Nature communications 2021 <https://www.nature.com/articles/s41467-021-21788-y>).

While the method described here is valid and clearly useful, there are alternative frameworks out there for these kinds of questions. A formal quantitative comparison is probably beyond the scope of this paper, but it would be useful to see a discussion of the strengths and weaknesses of different existing approaches.

Thank you. In the discussion, we added a paragraph about the modelling approaches that have been used to analyse serological assays. We write

Recently, modelling approaches have greatly increased our understanding of cross-reactive serological assays. For instance, these approaches helped improve diagnostic based on the value of serological assays [20], used population level data to inform individual diagnostic [21], informed risk of infection based on antibody dynamics [22-24], quantified the responses due to multiple influenza infections to detect recent infections [25]. Transversal serological studies can also provide insights on key epidemiological parameters, such as the attack rate and the mode of circulation of the viruses, by combining information from individuals from many locations and sociodemographic backgrounds [14,26].

The paper by Hay et al. proposes a general approach to study cross-reactivity in assays. Conceptually, our approaches are very similar. However, we propose a set of assumptions that are more adequate for the analysis of VNT assays for CHIKV and ONNV. In particular, we chose in the model an antibody response model that assumed a zero-truncated Poisson distribution both for the homologous and for the cross-reactivity response. The models of Hay et al. assume a normal distribution of the response. Moreover, in our model the cross-reactive response is stochastic: individuals can develop, or not, a response after an infection. We discuss more specifically this article:

Our approach shares many similarities with the general approach of Hay et al. [24], which proposes a statistical framework to study very diverse cross-reactive serological assays. However, the specificity of the viruses and assays used here required that we developed a unique antibody response and cross-reactivity model. In particular, we chose a zero-truncated Poisson model for the response and we specified in our model that cross-reactivity happens only for a fraction of the infections (approximately 80% of the CHIKV infections and 20% of the ONNV infections) (Table 1).

Line 108: “independent of the response against the infecting virus “- what is the evidence for or against this assumption that cross reactivity is independent of infecting virus antibody titre boost? Intuitively one might expect more robust generalised responses to be more cross reactive.

Thank you for this interesting comment. We tested different models where the cross-reactivity term was not independent of the response and we couldn't find any model that improved the fit to the data. We now describe one model in the section “Model Comparison and Sensitivity analysis”:

In the baseline model, we assumed that the cross-reactive response was independent of the response against the infecting virus. We tested an additional model where the cross-reactive response was a ZTP with a parameter proportional to the observed response for the infecting virus. For instance, in the case of an infection by CHIKV, the probability density function of the cross-reactive ONNV response is

$$CR(n_j^o, n_j^c, \sigma^{c \rightarrow o}) = \text{Poisson}(n_j^o, \sigma^{c \rightarrow o} \cdot n_j^c + \varepsilon | n_j^o > 0),$$

where $\varepsilon = 0.01$ is a small term that was added to ensure that the parameter of the Poisson distribution is non-zero. The probability density function for the VNT of the two viruses is

$$P(n_j^c, n_j^o | I_j = (1,0), \theta, X_j) = \text{Response}(n_j^c, \sigma^c) \times [p^{c \rightarrow o} CR(n_j^o, n_j^c, \sigma^{c \rightarrow o}) + (1 - p^{c \rightarrow o}) \delta(n_j^o)].$$

The value of the DIC does not show a stronger support for this model than for the baseline model (DIC = 3089 versus 3092 for the baseline model).

The baseline model and the model where the responses are proportional have very similar interpretations. We extended the Supplementary table 3 with the parameter estimates for this model, which shows that the CHIKV and ONNV prevalence are the same for both models (models 1 and 3 here). We also added the Supplementary Figure 4 that shows that the distribution of the responses upon infection are similar.

Supplementary Table 3. Parameter estimates for the baseline model (Model 1), the model with one-way cross-reactivity where anti-ONNV antibodies never induce a CHIKV response (Model 2), and the model where the cross-reactive response is proportional to the response for the infecting virus (Model 3). The numbers provided are the mean and 95% credible interval.

Parameter	Description	Model 1	Model 2	Model 3
σ^C	CHIKV response to CHIKV infection	2.4 (2.1 – 2.6)	2.5 (2.3 – 2.7)	2.4 (2.2 - 2.7)
σ^O	ONNV response to ONNV infection	4.0 (3.6 – 4.3)	3.9 (3.6 – 4.3)	3.9 (3.6 - 4.2)
$\sigma^{C \rightarrow O}$	ONNV response to CHIKV infection	2.0 (1.7 – 2.3)	2.3 (2.0 – 2.5)	1.2 (1.2 - 1.3)
$\sigma^{O \rightarrow C}$	CHIKV response to ONNV infection	2.9 (2.1 – 4.3)	-	1.3 (1.2 - 1.3)
p^C	CHIKV prevalence in Mali	0.13 (0.09 – 0.18)	0.19 (0.16 – 0.22)	0.11 (0.07 - 0.16)
p^O	ONNV prevalence in Mali	0.29 (0.25 – 0.34)	0.26 (0.23 – 0.29)	0.31 (0.27 – 0.35)
$p^{C \rightarrow O}$	Fraction of CHIKV infected that have a positive ONNV response	0.80 (0.72 – 0.87)	0.84 (0.78 – 0.89)	0.78 (0.68 - 0.85)
$p^{O \rightarrow C}$	Fraction of ONNV infected that have a positive CHIKV response	0.22 (0.08 – 0.33)	0	0.29 (0.16 - 0.39)
$f_1^C(tropical)$	Risk of CHIKV infection in tropical vs semi-arid regions	0.85 (0.52 – 1.33)	0.87 (0.63 – 1.17)	0.88 (0.48 - 1.51)
$f_1^O(tropical)$	Risk of ONNV infection in tropical vs semi-arid regions	0.77 (0.58 – 1.0)	0.74 (0.56 – 0.94)	0.77 (0.59 – 1.0)
$f_2^C(female)$	Risk of CHIKV infection for females vs males	1.9 (1.1 – 2.9)	1.8 (1.2 – 2.4)	1.7 (0.81 - 2.9)
$f_2^O(female)$	Risk of ONNV infection for females vs males	1.1 (0.80 – 1.5)	1.0 (0.73 – 1.4)	1.2 (0.87 - 1.6)

Supplementary Figure 4. Comparison of the response to infection for different models of cross-reactivity. Probability distribution of CHIKV response (A, D) and ONNV response (B,C) upon a CHIKV infection (top row) and an ONNV infection (bottom row). The distributions are presented for the baseline model and for the model where the cross-reactive response is proportional to antibody titer boost of the infecting virus.

We also commented our results in the section *Sensitivity analysis*:

We also tested a model where the cross-reactive response was proportional to the infecting antibody titer boost of the infecting virus. The DIC of this model and the baseline model were similar, showing that these models explained the data equally well (see Methods section Sensitivity analysis: The cross-reactivity model). The interpretation of results in terms of prevalence and response parameters was very similar (Supplementary Table 3 and Supplementary Figure 4). Complementary studies would be needed to distinguish between these two models and understand the dynamics of the antibody response in detail.

Minor comments:

Add equivocal results to table 1?- surely part of the benefit of the model-based classification is the reduced equivocal classification? It would also be helpful to make it clear what the total number of sera per site are in this table.

Thank you for the suggestion. We added the number of equivocal classifications. We also added the following table containing the demographic characteristics in the supplementary material:

Sampling site		Participants	Males	Females	Median age, y
North	Diema	109	27	82	28
	Bandiagara	187	58	129	35
	Niono	65	9	56	35
South	Bamako	129	50	79	32
	Kadiolo	136	40	96	30
	Kita	40	16	24	23
	Bougouni	127	42	85	45
Total		793	242	551	33

Figure 4 legend- “E-F” -> “E-H”

Done

Line 267: “the classical method was also more accurate at classifying ONNV infections” – do you mean model based method here- both look good, but model slightly better?

Thank you. Indeed we meant that both methods were good but the model-based classification had higher performance. We have rephrased:

In the scenario of 100% ONNV prevalence and 0% CHIKV, the classical method was very accurate with the classification of ONNV infections (Fig. 4F): ONNV infections were correctly classified as ONNV positive in 90% (95% CrI: 85% - 97%) of the cases and the probability that they were classified as CHIKV positive was only 2% (95% CrI: 0% - 3%).

No risk factor analysis for Martinique (i.e. equivalent of fig 3)?

Unfortunately, we did not have access to the age, sex or sampling locations of the participants in Martinique.

Check equation 5 and 6- p -> sigma?

The parameter p actually describes the probability of occurrence of a cross-reactive response, while sigma is the intensity of the response. To make this difference more explicit we now write sigma as a parameter of the function *Response*. We also mention sigma in the cross reactivity function *CR*.

REVIEWERS' COMMENTS

Reviewer #2 (Remarks to the Author):

The authors have appropriately addressed all of my comments